# Cenozoic Fault Growth Mechanisms in the Outer Apulian Platform

**Fabrizio Agosta [1,2,*], Angela Vita Petrullo [2,3], Vincenzo La Bruna [4] and Giacomo Prosser [1]**

[1]  Department of Sciences, University of Basilicata, Via dell'Ateten Lucano 10, 85100 Potenza, Italy
[2]  Geosmart Italia S.r.l.s., Via Di Giura 54, 85100 Potenza, Italy
[3]  Agenzia per la Protezione Ambientale della Basilicata, 75100 Matera, Italy
[4]  Department of Geophysics, Federal University of Rio Grande do Norte, Natal 1524, RN, Brazil
[*]  Correspondence: fabrizio.agosta@unibas.it

**Abstract:** This work focuses on a ca. 55 km-long extensional fault zone buried underneath the foredeep deposits of the southern Apennines, Italy, with the goal of deciphering the Cenozoic fault growth mechanisms in the Outer Apulian Platform. By considering public 2D seismic reflection profiles, well logs, and isochron maps data, the study normal fault zone is interpreted as made up of four individual fault segments crosscutting Top Cretaceous, Top Eocene, Top Miocene, and Top Pliocene chrono-stratigraphic surfaces. The computed cumulative throw profiles form either bell-shaped or flat-shaped geometries along portions of the single fault segments. The computed incremental throw profiles also show an initial fault segmentation not corresponding with the present-day structural configuration. Data are consistent with the initial, post-Cretaceous fault segments coalescing together during Miocene–Pliocene deformation and with fault linkage processes localizing at the stepover/relay zones. Pleistocene faulting determined the evolution of a coherent fault system. The computed *n*-values obtained for the single time intervals by considering the maximum fault throw–fault length relations indicate that the fault segments formed scale-dependent geometries. Variations of these computed values are interpreted as due to the higher degree of maturity reached by the entire fault system during Miocene to Pleistocene deformation.

**Keywords:** fault throw; fault linkage processes; stepover/relay zones; normal faults; Adria Plate; central Mediterranean

## 1. Introduction

The Apulian Platform is one of the largest Mesozoic carbonate platforms of the central Mediterranean area, forming the outcropping portion of the relatively rigid continental Adria microplate surrounded by Tertiary–Quaternary collisional fold-and-thrust belts, ftb [1–5]. In particular, the Apulian Platform is partly buried beneath the foreland basins, respectively, associated with the E- to NE-verging Apennines ftb, westward, and to the SW-verging Dinaric–Hellenic ftb, eastward [6,7]. Although it is commonly considered a slight deformation structural domain with respect to the flanking ftb's, the structural setting of the Apulian Platform is rather complicated as it represents a zone of interaction between the neighboring opposite-verging chains. The kinematic models proposed for the whole central Mediterranean area consider the Apulian Platform as part of the Adria microplate [8–10], which was dissected since Late Cretaceous by a major, left-lateral tectonic lineament cross-cutting the whole continental lithosphere [11–13]. Within the Apulian Platform, the presence of Upper Cretaceous—Eocene, NE-trending structural grabens [14] were interpreted by Vitale et al. [15] as due to an abortive rift system Cretaceous–Paleocene in age. More recently, following the work of Korneva et al. [16], Laurita et al. [17], and Panza et al. [18,19], Agosta et al. [20] documented the occurrence of Cretaceous transtensional tectonics by investigating the structural control exerted by high-angle faults bounding the bauxite deposits exposed in the Murge Plateau of southern Italy.

Since the Cenozoic sedimentary succession is not widely exposed at the surface in the forebulge area of the Murge Plateau [21], whereas it is quite common westward where the Apulian Platform plunges below the Pliocene–Pleistocene infill of the Bradano Trough foredeep basin [22]. For this reason, previous authors employed the public seismic reflection, well log, and isochron datasets (VIDEPI, https://www.videpi.com/videpi/videpi.asp, accessed on 13 November 2020) to assess both geometry and distribution of the Eocene and Miocene deposits topping the Mesozoic carbonates [23]. As a result, the latter authors were able to decipher the 3D distribution of the buried Tertiary deposits and also documented the occurrence of Upper Messinan–Lower Pliocene, oblique-slip normal faulting. Focusing on the transtensional deformation that took place prior to the Plio-Quaternary extension of the Apulian Platform [24,25], in this work, we aim to unravel the Cenozoic growth mechanisms of a large-scale fault zone crosscutting the Outer Apulian Platform. We investigate an about 50 km-long, high-angle fault zone buried below the Bradano Trough foredeep basin of southern Italy and discuss its time-dependent growth mechanisms in light of the existing bibliography.

In more detail, we focus on the analysis of both cumulative and incremental fault throw variations computed for the individual segments forming the study, ca. 50 km-long, high-angle fault zone. We first discuss this data in light of the extensive work published on the process of fault growth by linkage of pre-existing fault segments [26–33] to assess their scale-independent [34] or scale-dependent geometries [35]. Then, we compare the results with those recently published by La Bruna et al. [36] for the high-angle fault zones displacing the Inner Apulian Platform along the axial zone of the southern Apennines ftb, Italy. As a result, we shed new light on the Cenozoic tectonic evolution of the Outer Apulian Platform prior to the Plio-Quaternary foreland bulging, which profoundly affected its present-day structural setting.

## 2. Geology of the Study Area

The Apulian Platform currently crops out in the Gargano, Murge, Salento, and Monte Alpi areas of southern Italy (Figure 1). There, it is commonly subdivided into two main domains labeled as Inner and Outer Apulian Platforms, respectively (Figure 1). The Outer Apulian Platform forms part of the present-day foreland domain of the southern Apennines ftb, which developed due to E- to NE-verging thrusting of Meso-Cenozoic, shallow- and deep-water sedimentary sequences [37–42]. The Apulian Platform was involved in rapid subsidence due to its eastward rollback, favoring the formation of deep foredeep basins and thrust-sheet top basins on the allochtonous units of the ftb. Concomitant with the progressive piggy-back propagation of the thrust sheets, these foredeep and thrust-top basins migrated eastwards during the Miocene–Pleistocene tectonic evolution of the ftb [43,44]. During the same time interval, the Outer Apulian Platform was partly involved in the formation of the southern Apennines and the Dinaric ftb's, and was affected by local-scale strike-slip faulting [3,45–47]. Deep-to-shallow water siliciclastic deposits topped the Outer Apulian Platform and infilled the Bradano Trough that formed the Late Pliocene–Middle Pleistocene foredeep basin of the southern Apennines ftb [40,48,49].

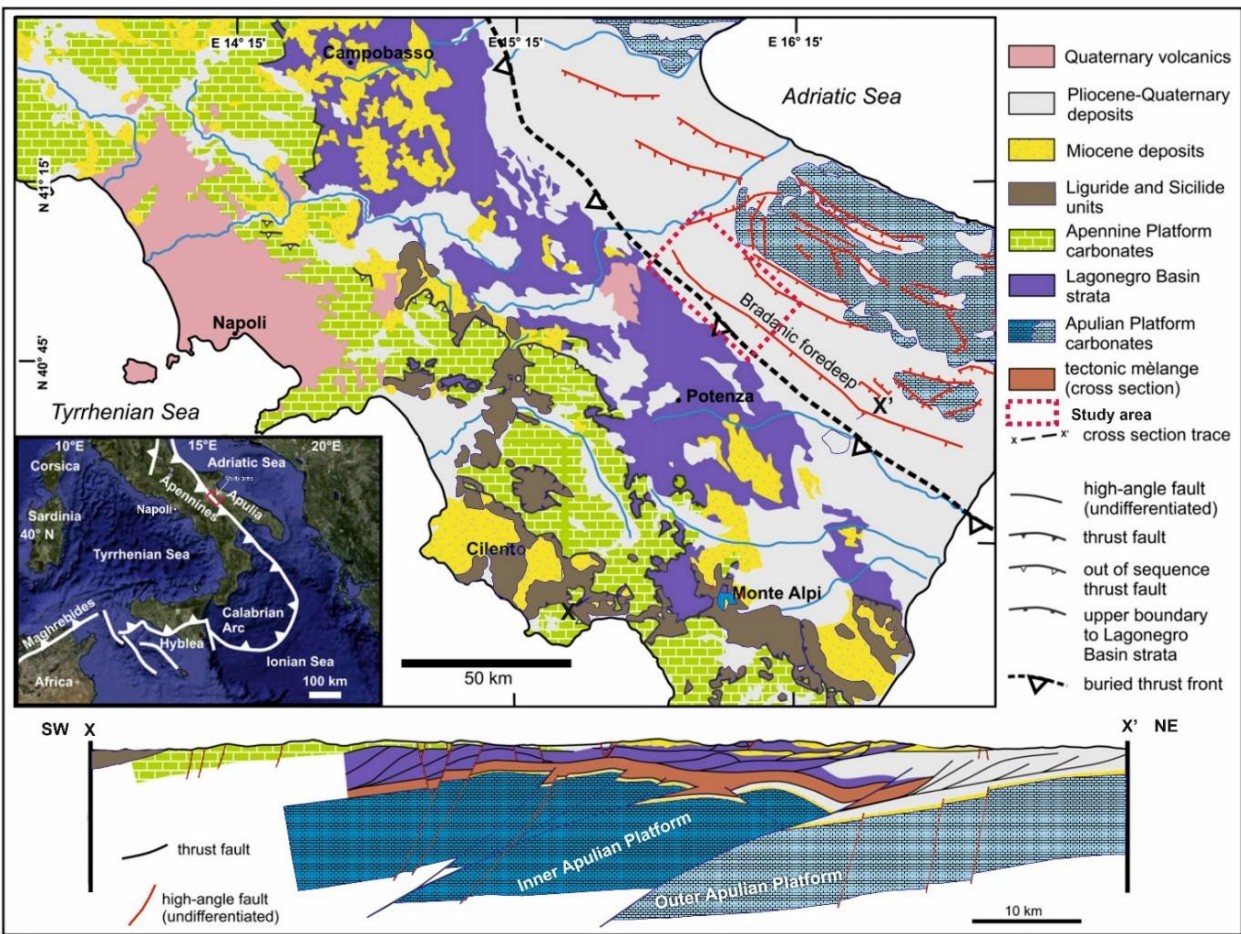

**Figure 1.** Simplified geological map of southern Apennines, Italy, and NE–SW geological cross-section modified from [50]. The sketch map in the inset shows the structural setting of southern Italy, in which the red box represents the location of the study area (also reported in the main figure).

### 2.1. Stratigraphy

The Outer Apulian Platform is made up of rocks that originated in a variety of depositional settings, from evaporitic basins to open carbonate platform environments (Figure 2). On top of an unknown Variscan basement, this platform includes Permian–Triassic, alluvial, terrigenous red beds similar to the Verrucano Fm., and Upper Triassic evaporites and dolomites of the Burano Fm. [51,52]. The buried Jurassic succession consists of indistinct shallow-water carbonates, whereas the exposed Cretaceous carbonates form two main stratigraphic units, the Valanginian–Cenomanian Calcare di Bari Fm., and the Coniacian–Early Campanian Altamura Fm., respectively [21,53].

Starting from the Paleogene, the whole Apulian Platform underwent uplift, sub-aerial exposure, and formation of widespread unconformities on top of the Cretaceous carbonates [54]. In most places, the oldest Tertiary deposits consist of Eocene limestone with Nummulites (Calcari a Nummuliti Fm.) and of calcareous breccia (Brecce di Lavello Fm.), which locally include tuffs and basalts forming either distinct levels or clasts [55–57]. Lacking the Oligocene interval due to a regional-scale regression, the Eocene rocks were topped by Lower Miocene calcareous breccia (San Ferdinando Fm.) and by Middle Miocene carbonate packstones/wackestones (Bolognano Fm.) originally deposited in a carbonate shelf environment with significant terrigenous inputs [56,58]. During Messinian, foredeep and evaporitic basins topped both western and eastern portions of the whole Apulian Platform [56,59,60].

Generally, the Inner Apulian platform buried below the allochtonous units of the southern Apennines fold-and-thrust belt, ftb, was characterized by a quite homogenous

sedimentary succession. The occurrence of a Late Cretaceous deepening-upward trend was recorded by local-scale pelagic sedimentation [61]. Westward, in an area extending close to the Monte Alpi massif (cf. Figure 1), the topmost portion of the Inner Apulian platform consists of Aptian-to-Turonian/Cenomanian limestones, which are unconformably topped by Messinian rocks [62,63].

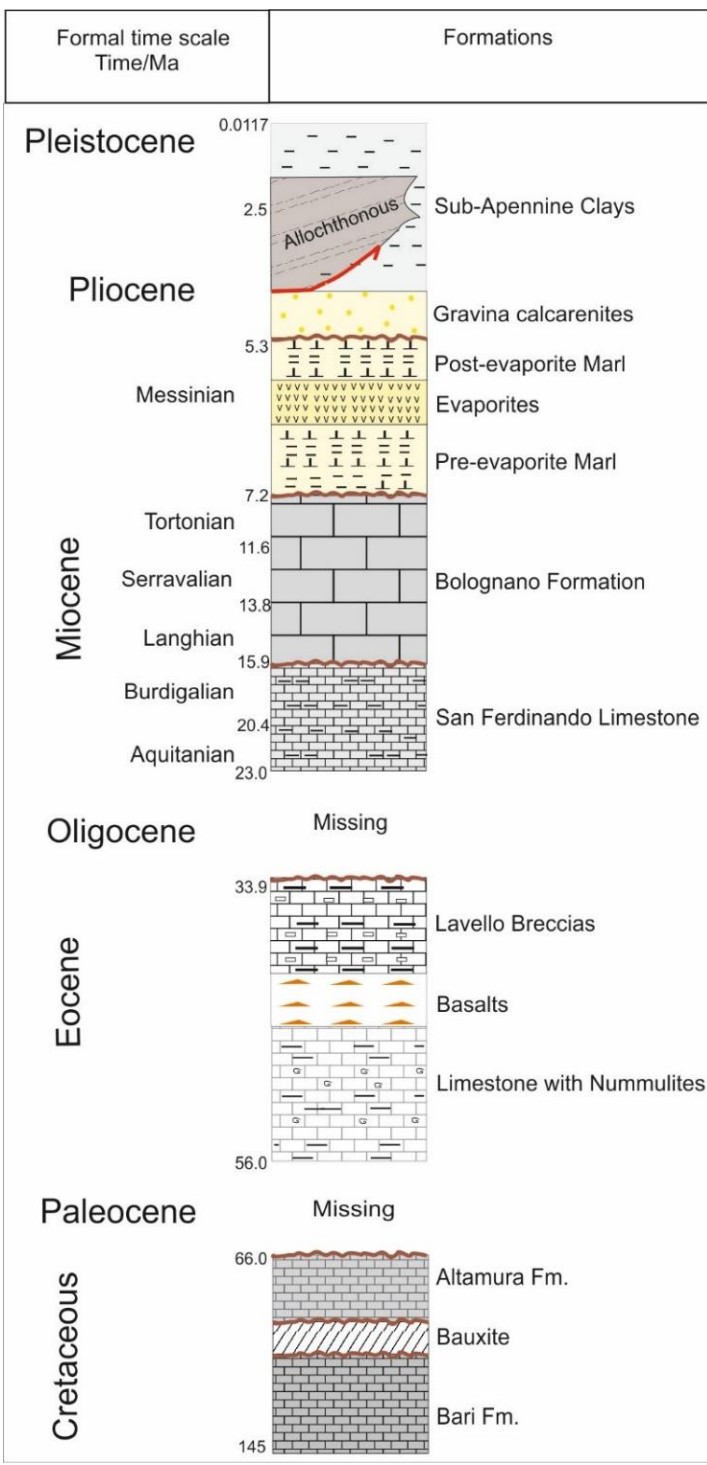

**Figure 2.** General stratigraphic column (not to scale) recognized in exploration wells showing the succession of the Apulian Platform and the foredeep units above.

During the Lower Pliocene–Early Pleistocene, open marine conditions persisted throughout the whole Apulian Platform domain as a consequence of the evolution of the Southern Apennines foredeep basin [56]. To the east, the basin floor sandstone lobes form a turbidite complex representing the first sedimentary infill of the Bradano Trough. Local structural highs of the foredeep were characterized by shallow-water carbonate sedimentation of the Calcarenite di Gravina Fm. [64]. The Early Pleistocene mudstones and silty-clay hemipelagic deposits of the Argille subappennine Fm. were topped by shallow marine sandstones and fluvio-deltaic conglomerates, which were deposited during a general regressive stage [65]. Widespread marine and/or continental terraces were deposited during the final uplift of the foredeep.

### 2.2. Structural Setting

From west, Tyrrhenian coastline, to east, Adriatic coastlines, the southern Apennines ftb consists of a series of thrust sheets including: (i) terrigeneous and siliceous rocks of the Liguride/Sicilide oceanic basin; (ii) shallow-water carbonates of the Apenninic Platform; (iii) deep-water mixed terrigenous-carbonate rocks and siliceous cover of the Lagonegro Basin; (iv) carbonates of the Apulian Platform (Figure 1). The study area is located in the foredeep-foreland domains of the southern Apennines ftb, in between the Vulture Volcano and the Murge Plateau of southern Italy. There, the western edge of the Outer Apulian Platform is buried underneath the Pliocene foredeep deposits and tectonically covered by the frontal portion of the Southern Apennines ftb.

Thrusting of the outermost portion of southern Apennines ftb involved a complex tectonic assemblage of Mesozoic and Tertiary sedimentary units [40]. The thrust sheets overrode Early–Middle Pliocene clastic units topping the Apulian Platform and the western sector of the Bradano Trough [40,56,66]. The Inner Apulian Platform [40,67] was affected by Late Pliocene–Early Pleistocene thick-skinned thrusting [49,68]. Differently, at the same time, the Outer Apulian Platform was affected by a significant bulging and development of a Plio-Pleistocene foreland basin system [21,23,69].

The present-day structural configuration of the Murge Plateau includes NW–SE trending normal faults bounding tectonic grabens, which are infilled with Plio-Pleistocene sedimentary successions [70]. Minor sets of Quaternary normal faults trending ca. NE–SW, E–W, N–S, and NNW–SSE were documented all over the Murge Plateau [71,72]. The NW–SE trending normal faults likely reactivated Mesozoic [16,20,21,73], and Miocene high-angle faults [71,74–76]. The major NW–SE faults are made up of individual segments juxtaposing stratigraphic units of different ages against each other. Previous work on a ca. $2 \times 10$ km$^2$-wide, foredeep-foreland transitional area conducted by Petrullo et al. [23] documented the regional-scale structural architecture of the Outer Apulian Platform (Figure 3a). The latter authors inferred that Messinian–Early Pliocene transtensional faulting was associated with the formation of small pull-apart basins, which were bounded by NNW–SSE trending splay faults localized at the releasing jogs of interacting NW–SE faults, and that Early Pliocene–Late Pleistocene faulting was characterized by dip-slip extensional kinematics.

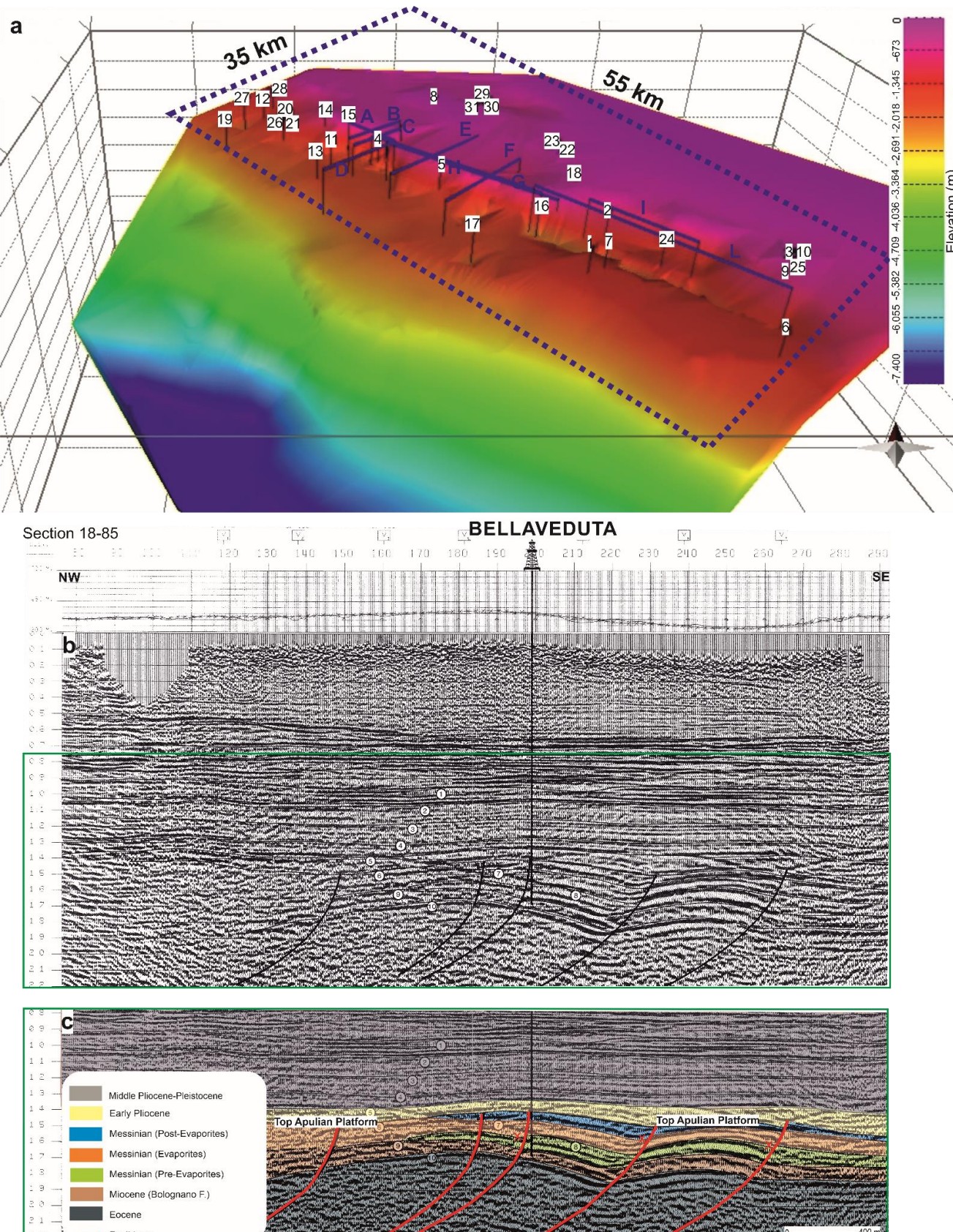

**Figure 3.** (**a**) Isobath map of top Apula surface (Upper Cretaceous to Messinian age) obtained after 2D seismic reflection and well log data interpretation, and by taking into account the isobath map by Nicolai and Gambini [77]. The color bar on the right shows the surface elevation in meters with

different colors. The dashed blue line serves to identify the investigated area. (Wells: 1-Agatiello1, 2-Arcieri, 3-Banzi, 4-Bellaveduta, 5-Calvino, 6-Donna Caterina, 7-Forenza2, 8-Gaudiano, 9-Genzano1, 10-Genzano2, 11-Lavello1, 12-Lavello2, 13-Lavello5, 14-Lavello4, 15-Lavello6, 16-Maschito1, 17-Maschito2, 18-Matinella1, 19-Melfi1, 20-Masseria Spavento1, 21-Masseria Spavento3, 22-Montemilone1, 23-Montemilone2, 24-Musacchio1, 25-Muscillo1, 26-Rendina1, 27-San Raffaele1Dir, 28-Serra Spavento6, 29-SpgnolettiB, 30-SpagnolettiC, 31-SpagnolettiD. Seismic sections: A-Pz 18-85, B-14-85, C-15-85, D-16-85, E-VN4, F-VN5, G-VN1, H-483-82, I-440-81, L-384-79). (**b**) Not-interpreted and (**c**) interpreted NW–SE oriented seismic reflection profiles 18-85 across the northwestern sector of the study area, showing the listric geometry of the N–S directed fault zone that displaces gently folded Eocene–Miocene succession (4-calcarenitic level, 5-top Early Pliocene, 6-top carbonates, 7-gypsum, 8-top lacustrine, 9-top limestone and dolomite, 10-top Cretaceous). See (**a**) for location.

## 3. Methodology

The 3D geological modeling of the subsurface rock volumes lying underneath the area in between the Vulture Mt. and the Murge Plateau, southern Italy, was carried out by considering the 2D seismic reflection profiles, well logs, and isochron maps available from the VIDEPI website (Figure 3a). Data were interpreted according to lateral continuity of the main seismic reflectors in light of the stratigraphic information extrapolated by well-log analysis (Figure 3b,c). Furthermore, the 3D geological modeling performed by using the Move® software version 2015.1 was carried out by integrating data from the aforementioned subsurface data with the published Top Apula isobath map [77]. Seismic profiles and well logs were imported in Move™ for time-to-depth conversion and also for digitization of the main stratigraphic horizons. Specifically, faults and horizons were interpreted on the seismic sections. All available well data were also uploaded into the project to further constrain our interpretation. In order to better constrain our interpretation, we merely focus on the seismic lines passing through or in the vicinity of existing wells (Figure 3a). Afterward, specific velocity values were attributed to the stratigraphic units in order to perform a time-to-depth conversion. The velocity value of single horizons identified in the seismic sections was calculated with the "direct conversion time-depth method", which considered the thickness of individual units recognized in well logs (Z) and the corresponding TWT on the seismic section. This method employs a constant velocity for the single layers and does not take into account any possible lateral variation of the stratigraphic units. The average velocity of single layers, taking into account the correlation between deep well-logs and seismic section, was hence calculated thanks to the equation: Velocity = Z/(TWT/2). Hereafter, we report the average velocity values calculated and then used for the time-to-depth conversion:

- Late Pliocene (Conglomerates) 1781 m/s;
- Middle Pliocene (Sands) 2763 m/s;
- Middle Pliocene (Clay) 3112 m/s;
- Middle Pliocene (Calcarenites) 2949 m/s;
- Early Pliocene 2599 m/s;
- Evaporitic Marl (Messinian) 2621 m/s.

In detail, the faults in 2D are considered idealized planes forming cut-off lines with individual chronostratigraphic surfaces. The vertical separation of single cut-off lines is then computed in order to assess both cumulative and incremental throw profiles. Furthermore, the measured values of maximum fault throw, $D_{max}$, and individual fault segment length, L, to compute the *n*-value by employing the following equation:

$$D_{max} = cL^n \tag{1}$$

where the *c* value is associated with the rock's mechanical properties [78–81]. Further details are reported in the following text.

## 4. Results

Three-dimensional geological modeling of the indistinct Top Apulian carbonate surface, representing the base of the Pliocene foredeep deposits (Figure 4). We note that the indistinct Top Apulian carbonate surface is not isochronous due to the major unconformity present at the base of the Pliocene deposits. Consequently, its interpreted age spans from Upper Cretaceous to Messinian [58]. As already reported by Petrullo et al. [23], this surface is crosscut by two sub-parallel, NW–SE trending and SW-dipping escarpments (Figure 5a,b), respectively representing major and minor morphological scarps.

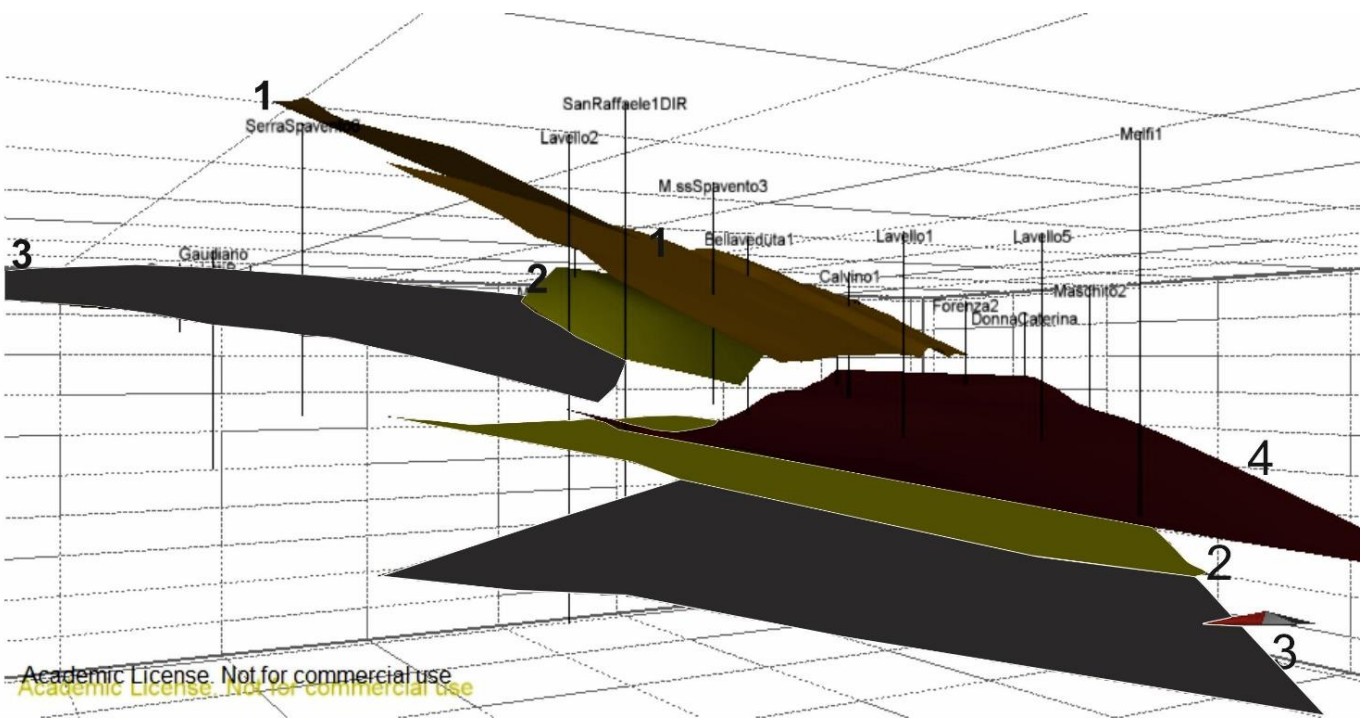

**Figure 4.** Marker surfaces reconstructed in hanging wall and footwall blocks offset by an NW–SE trending fault. The markers are numbered as follows: 1: Top Pliocene, 2: Top Eocene, 3: Top Cretaceous, 4: Top Miocene.

### 4.1. Fault Scarp Geometry

This study deals with the major NW–SE trending escarpment of the modeled indistinct Top Apulian carbonate surface (Figure 5a). This escarpment is ca. 1 km-high, 55 km-long, dips toward the SW, and downdrops southwestward the Eocene-to-Early Pliocene sedimentary succession. We recognize four main fault segments, which are respectively labeled as Fault A, Fault B, Fault C, and Fault D (Figure 5b). The individual fault segments show an average dip angle of ca. 50 degrees, as obtained from time-to-depth converted seismic profiles. In a plan view (Figure 5c), the NW–SE trending Fault A and Fault C form a right-stepping geometry and are linked together by the NNW–SSE trending Fault B. Differently, still, in a plan view, the NW–SE striking Fault C and Fault D are almost aligned one another, forming a slightly visible left-stepping geometry (Figure 5c).

In detail, observing the NE–SW oriented seismic reflection profile of Figure 6a, we note that Fault A bounds eastward the wedge-shaped Messinian units, including the pre-, syn-, and post-evaporitic sedimentary successions. We also note that some of the NW–SE trending minor faults located at the hanging wall of Fault A are topped by Early Pliocene clastic sediments (cf. first three fault traces on the left in Figure 6b). Differently, two sub-parallel normal faults with a listric geometry crosscut the footwall block of Fault A, displacing both Early Pliocene and Late Pliocene–Early Pleistocene stratigraphic units. The greatest thickness of the Messinian deposits is documented in correspondence of

the Bellaveduta well (Figure 6b), which lies in between the right-stepping Fault A and Fault C. Accordingly, Petrullo et al. [23] interpreted this thick Messinian succession as the infill of subsiding blocks located within the releasing jog, and hence assessed right-lateral, oblique-slip extensional faulting to the latter fault segments.

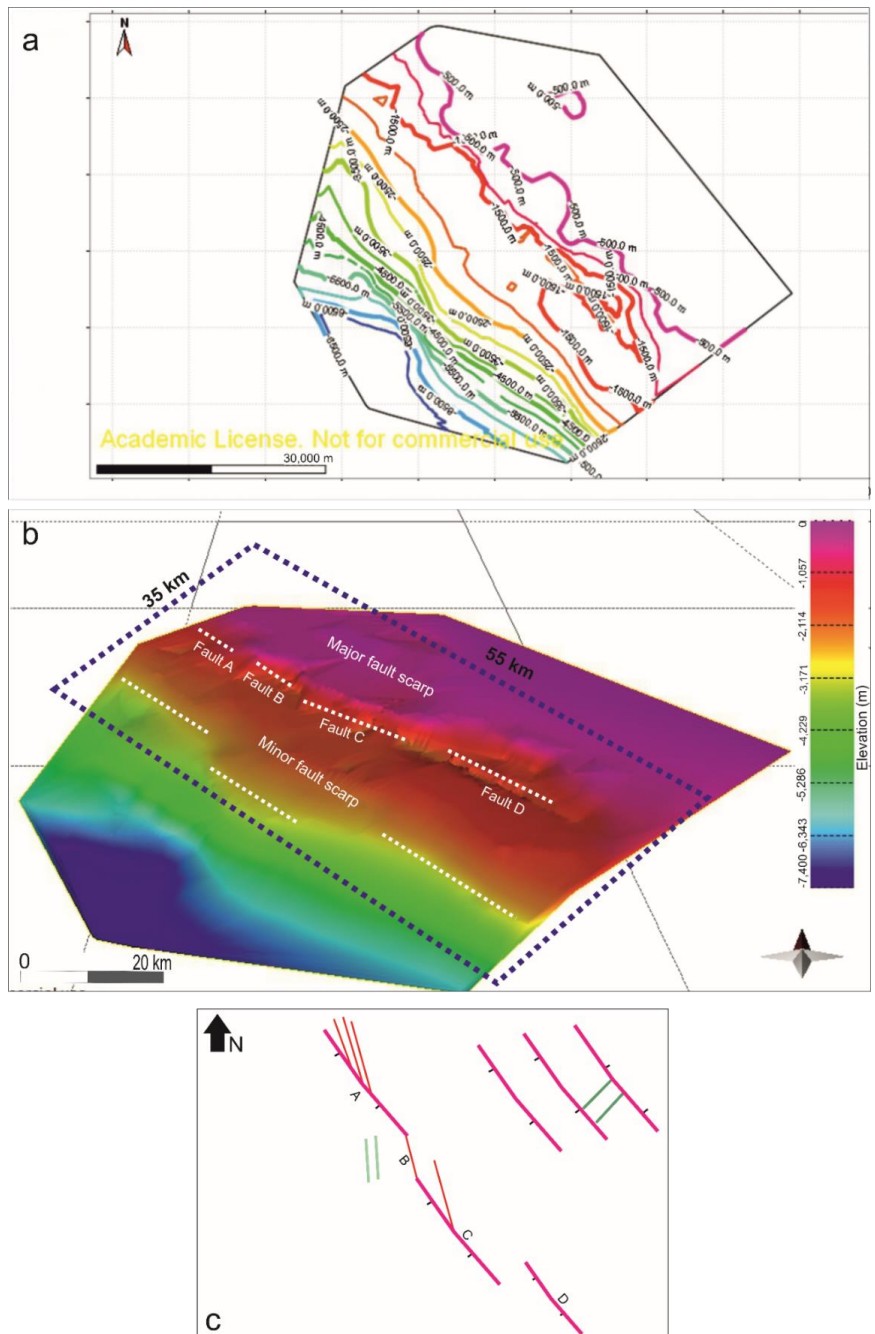

**Figure 5.** (**a**) Two-dimensional isobath map of Apulian Carbonates Platform. The isobath (every 500 m) are referred to the Apula top surface (Upper Cretaceous to Messinian age); (**b**) three-dimensional view of the top Apula surface, showing the architecture of the major fault scarp obtained by 3D modeling. The legend bar to the right shows with different colors the surface elevation in meters. The dashed blue line represents the investigated area. The isobath map was obtained after interpretation of 2D seismic reflection profiles, well log data, and by considering the isobath map by Nicolai and Gambini [78]); (**c**) present-day fault architecture shown in plan view. Traces of minor fault segments are also shown. The traces of the main fault segments are reported in magenta, red and green lines represent the traces of subsidiary high-angle faults.

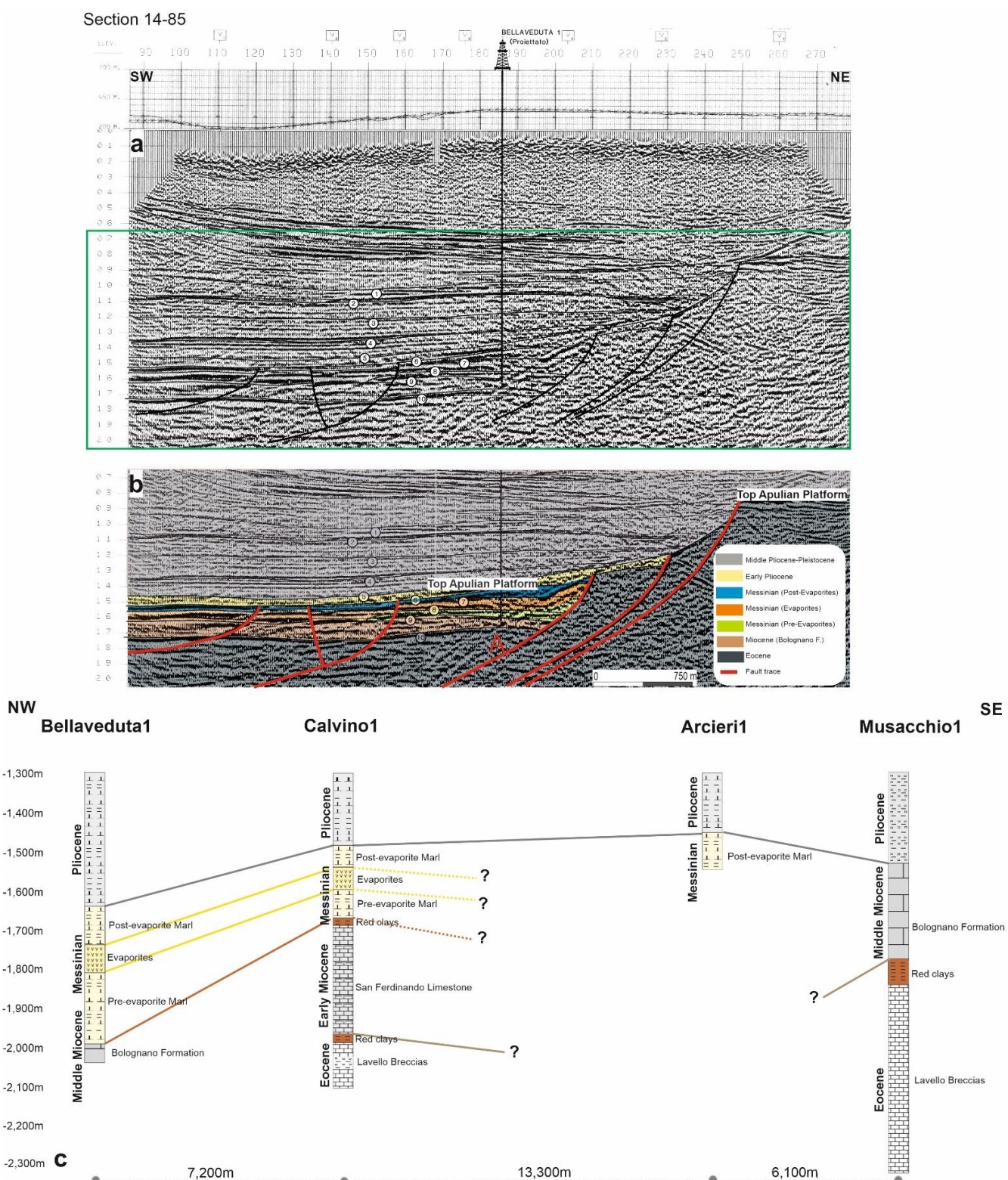

**Figure 6.** (**a**) Uninterpreted and (**b**) interpreted NE–SW oriented seismic reflection profiles 14-85, showing the listric geometry of the NW–SE trending faults, with syn-sedimentary activity during deposition of the Messinian deposits (1-conglomerates, 2-top sands of Bellaveduta well, 3-bottom sands of Bellaveduta well, 4-calcarenitic level, 5-top Early Pliocene, 6-top carbonates, 7-gypsum, 8-top lacustrine, 9-top limestone and dolomite, 10-top Cretaceous). (**c**) Schematic stratigraphic profile, parallel to the fault scarp, derived from the correlation of Bellaveduta1, Calvino1, Arcieri1, and Musacchio1 exploration wells. See Figure 3a for the location of the wells.

## 4.2. Thickness Profiles of Both Eocene and Pliocene Units

Focusing on the geometry of the Top-Pliocene, Top-Miocene, Top-Eocene, and Top-Cretaceous stratigraphic surfaces, we compute the thickness distribution of both Pliocene and Eocene sedimentary units. In Figure 7, we only report the 3D geometry computed for the latter three surfaces because the Top-Pliocene surface would cover the others. There, the 3D thickness distribution computed for the Miocene (Figure 7b) and Eocene stratigraphic intervals (Figure 7c) are reported. We note that the Miocene thickness was computed only at the fault hanging wall because the Top Miocene stratigraphic surface is not present at the fault footwalls. The thickness variations of both the Pliocene and Eocene units have then been projected on vertical sections parallel to the strikes of the A to D fault segments (Figure 8). Conversely, as reported above, the thickness variations of the Miocene deposits are not considered due to their lack at the fault segment footwalls.

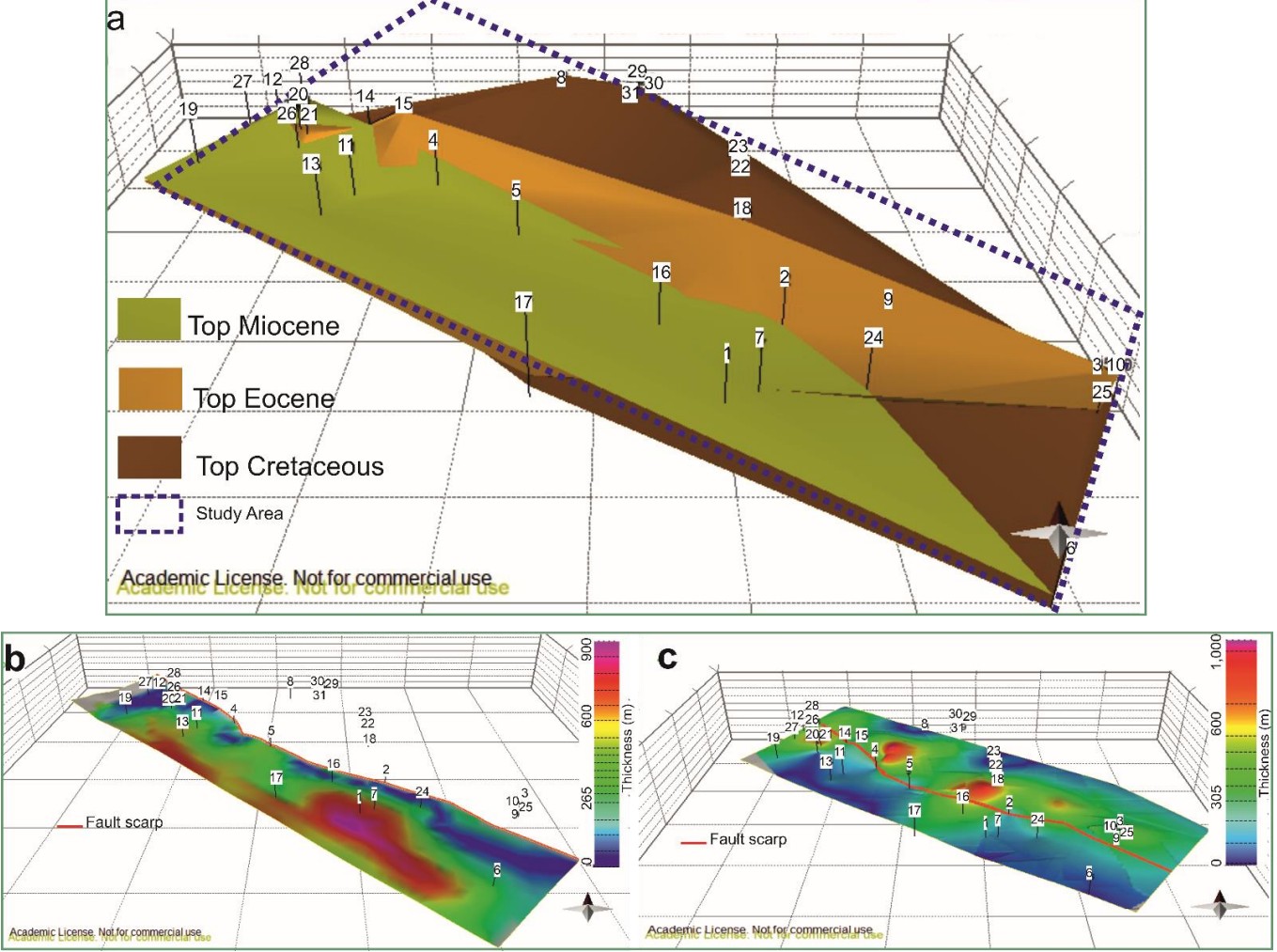

**Figure 7.** (**a**) Three-dimensional geometry of the Top Miocene, Top Eocene, and Top Cretaceous surfaces from well log data interpretation. The three surfaces form angular unconformities. (**b**) Thickness distribution of the Miocene stratigraphic interval was calculated considering both Top Miocene and Top Eocene surfaces. (**c**) Thickness of the Eocene stratigraphic interval calculated considering both Top Eocene and Top Cretaceous surfaces. In the latter figures, the trace of the main fault scarp identified in the isochron map (downloaded from the ViDEPI website) is shown.

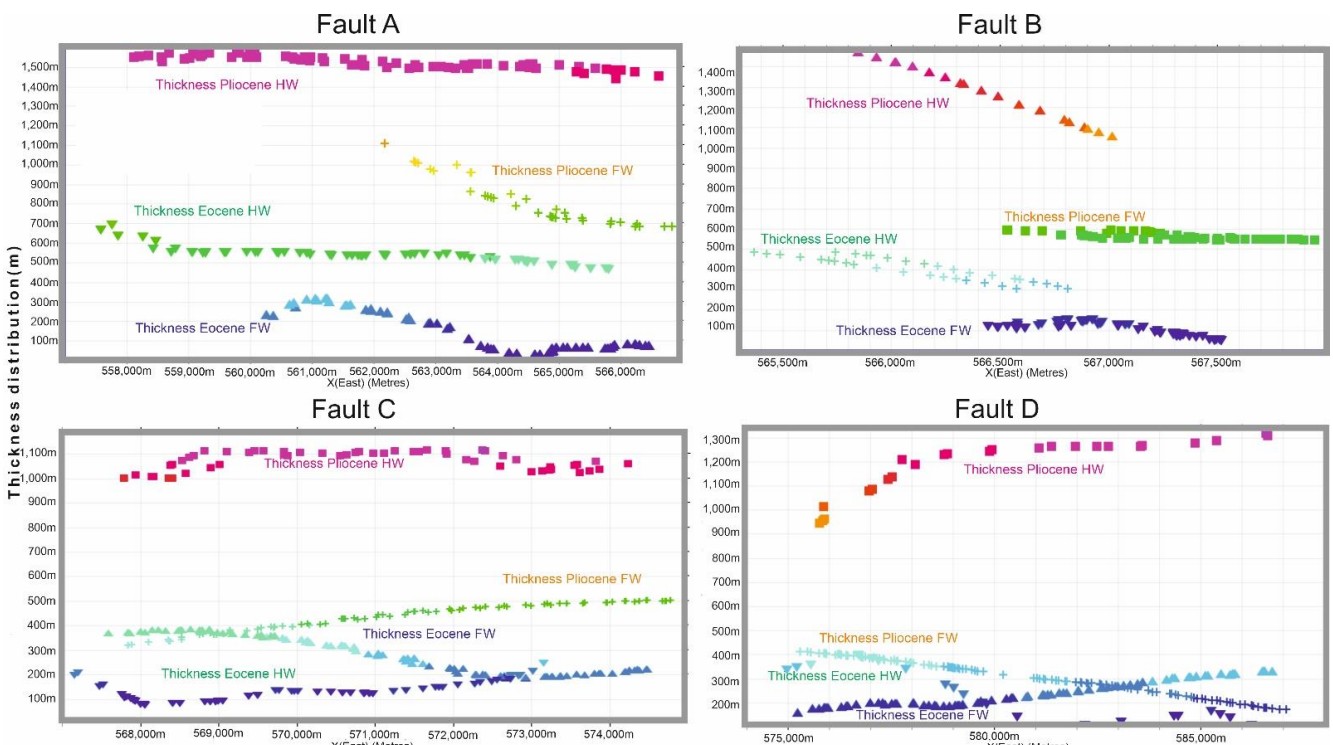

**Figure 8.** Thickness variations of Eocene and Pliocene stratigraphic units projected along sections oriented parallel to the strike of the A to D fault segments.

The Pliocene deposits show the maximum thickness, with values ranging from ca. 1 to 1.5 km at the fault hanging walls and from ca. 0.2 to 1.1 km at the fault footwalls (Figure 8). We note that the computed Pliocene thicknesses show nearly flat profiles along the A, C, and D fault segments. On the contrary, the Pliocene deposit shows greater thickness northward along Fault B, in the vicinity of Fault A. The Eocene deposits exhibit values of computed thickness ranging from ca. 0.1 to 0.7 km along the A, B, and D fault hanging walls and between ca. 0.1 and 0.4 km along their footwalls (Figure 8). Differently, along fault C, the Eocene deposit is thicker (ca. 0.4 km) at the fault footwall rather than at the hanging wall (ca. 0.1 km).

### 4.3. Cumulative Fault Throw Profiles

The cumulative throw profiles of the A to D fault segments are computed on the basis of the cut-off lines formed by the single chrono-stratigraphic surfaces (Figure 9). Due to missing Miocene deposits at the fault hanging walls, hereafter, we present the cumulative fault throw data computed for the Top Cretaceous, Top Eocene, and Top Pliocene surfaces. The Top Cretaceous shows a relative constant throw distribution across the whole investigated area of ca. 1.4 km at Fault A. Locally, the Top Cretaceous surface displays an asymmetric throw profile geometry, which forms a distinct bell-shaped profile along both A and D faults. An abrupt drop is documented along Fault B, and an almost flat profile along Fault C. The cumulative Eocene throw distribution is also quite asymmetric across the whole fault zone. Its maximum throw of ca. 1.2 km is located at the northwestern edge of Fault C, whereas its minimum value of ca. 0.4 km is at the northwestern edge of Fault A. Along Fault C, the fault throw value constantly decreases southward down to ca. 0.7 km at the intersection between the C and D fault segments and then constantly increases up to ca. 1.1 km along Fault D. The cumulative Pliocene throw distribution shows flat profiles with constant values of ca. 0.2 km along both Fault A and Fault C, almost null values along Fault B, and a bell-shaped profile with a local maximum at ca. 0.6 km along Fault D.

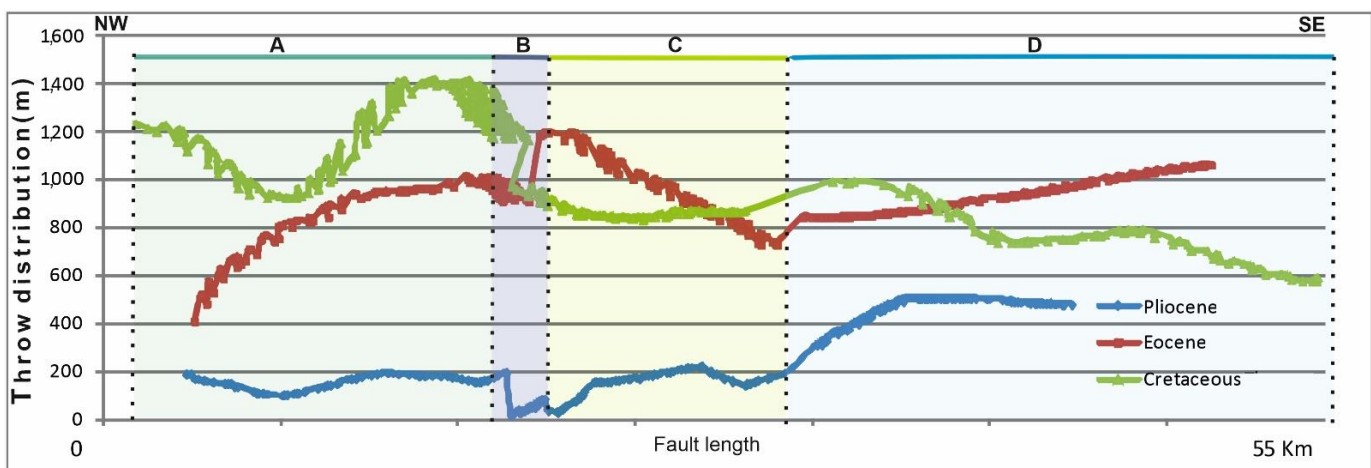

**Figure 9.** Total Top Pliocene, Top Eocene, and Top Cretaceous throw profiles computed for A to D fault segments.

*4.4. Incremental Fault Throw Profiles*

In order to assess the whole deformation history of the study regional-scale fault zone, we compute the throw increments related to the Paleocene–Eocene, Miocene–Pliocene and Pleistocene time intervals (Figure 10). The Paleocene–Eocene throw distribution is obtained by subtracting the cumulative Eocene and Pliocene throw values from the cumulative Cretaceous throw values. Similarly, the Miocene–Pliocene throw distribution is computed by subtracting the cumulative Pliocene throw values from the cumulative Eocene throw values. Of course, the Pleistocene throw values correspond to the cumulative ones. The maximum computed maximum vertical displacement ($D_{max}$)–Length (L) data for the whole analyzed fault segments are reported in Table 1.

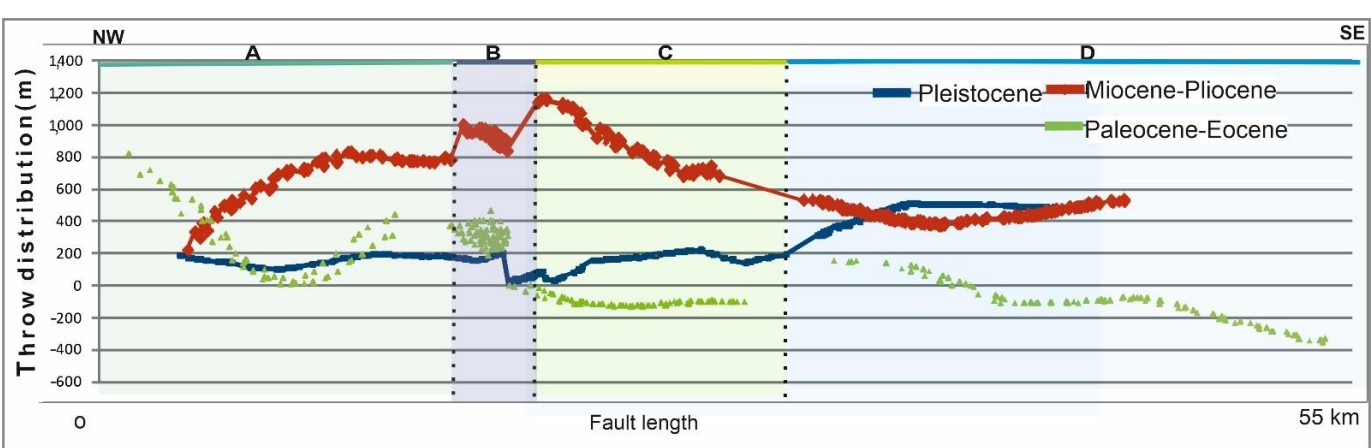

**Figure 10.** Pleistocene, Miocene–Pliocene, and Paleocene–Eocene throw profiles computed for A to D fault segments.

In detail, hereafter, we report the main results obtained for the three-time intervals:

- The Paleocene–Eocene throw distribution shows a maximum value of ca. 0.8 km at the northwestern tip of Fault A and local minima of ca. −0.4 km at the southern tip of Fault D. The negative throw value is related to the uplifted hanging wall block relative to the footwall block. Two separate bell-shaped profiles are documented along Fault A.
- The Miocene–Pliocene throw profile shows two separate bell-shaped profiles along the A to D fault segments. The first bell-shaped profile is characterized by a maximum value of ca. 1.2 km located at the intersection between Fault B and Fault C faults. The other maximum value of ca. 0.6 km is located at the southeastern end of Fault D.

- The Pleistocene throw distribution is very irregular, showing an asymmetric shape and numerous sudden variations. The computed values range from a few tens of meters (Fault A and Fault B) to ca. 500 m (Fault C and Fault D). Along the latter fault segments, the greatest values are located at their southeastern termination. Differently, the throw distribution profile is nearly flat along Fault A and almost null along Fault B.

**Table 1.** Name, length, and vertical throw of the computed fault segments related to the Paleocene–Eocene (green), Miocene–Pliocene (red), and Pleistocene (blue) fault segments.

| Paleocene–Eocene | | |
| --- | --- | --- |
| NAME | LENGTH (km) | THROW (km) |
| A1 | 9.0 | 0.83 |
| A2 + B | 10.0 | 0.49 |
| D1 | 6.5 | 0.18 |
| Miocene–Pliocene | | |
| NAME | LENGTH (km) | THROW (km) |
| A1 + A2 + B + C + D1 | 43.0 | 1.20 |
| D2 | 20.0 | 0.59 |
| Pleistocene | | |
| NAME | LENGTH (km) | THROW (km) |
| A1 + A2 + B | 16.0 | 0.22 |
| C + D1 + D2 | 23.0 | 0.53 |

## 5. Discussion

Results of 3D modeling of the buried Top Apula surface are interpreted to decipher the fault growth mechanisms producing the present-day structural architecture of the large-scale, high-angle extensional fault zone [23,56,77]. We recognize four individual fault segments, which are labeled as Fault A to Fault D from north to south, respectively, and displace the Top Cretaceous, Top Eocene, and Top Pliocene chrono-stratigraphic surfaces. After data interpretation and computation, both bell-shaped and flat-shaped cumulative throw profiles are documented (cf. Figure 9). The former throw profiles are assessed for Fault A and Fault D (Top Cretaceous), Fault A-to-Fault C (Top Eocene), and Fault C and Fault D (Top Pliocene). Differently, the flat-shaped profiles are assessed for Fault C (both Top Cretaceous and Top Pliocene) and Fault A (Top Pliocene). Overall, the aforementioned geometries of the cumulative throw profiles are interpreted as resulting from the growth of isolated fault segments, which likely interacted with one another during ongoing deformation [28,79,80,82–89]. We note that the NNW–SSE striking Fault B is characterized by sudden variations of both Top Cretaceous and Top Eocene cumulative throw profiles and by almost null Top Pliocene cumulative throw values. Along this portion, the abrupt throw variation (observed on the computed cumulative throw profile analysis) could be associated with a fault branch line [33].

Focusing on the computed incremental throw values, we now consider the single time intervals reported in Figure 10. The Paleocene–Eocene throw profile shows negative values along Fault C and for the southeastern portion of Fault D. We interpret these data as due to the widespread Late Cretaceous–Paleogene erosion of the Apulian Platform, which was associated with instability and emersion of the carbonates [52,90,91]. Such an erosion, if localized at the fault footwall, might have modified the geometry of the original chronostratigraphic surfaces and hence determined the apparent reverse motion along the aforementioned high-angle fault segments. Furthermore, on the basis of the incremental throw profiles, we classify the present-day fault zone into the following individual fault segments. Considering the dissection of the Top Cretaceous chrono-stratigraphic surface, we recognize the Paleocene–Eocene, isolated Fault A1, Fault A2 + B, and Fault D1 segments (Figure 11a). Differently, considering the dissection of the Top Eocene chrono-stratigraphic surface, we assess two single bell-shaped throw profiles for the Miocene–Pliocene time

interval (Figure 11b). Accordingly, these geometries are interpreted as due to the shearing of single fault segments labeled as Fault A1 + A2 + B + C + D1 and Fault D2, respectively. Differently, at that time, Fault D2 formed as an isolated fault segment. Considering the Pleistocene time interval, the computed throw profile is consistent with the complete linkage among the individual fault segments. Two distinct segments labeled as Fault A1 + A2 + B and Fault C + D1 + D2 are recognized (Figure 11c). This structural configuration hence shows that the whole study fault zone behaved as a coherent fault system during this time span. We note that Fault B pertained to the Fault A1 + A2 + B + C + D1 segment during Miocene–Pliocene and to the Fault A + A2 + B during Pleistocene. Accordingly, we interpret that Fault B acted as a soft linkage zone between Fault A2 and Fault C during fault evolution [26,92,93].

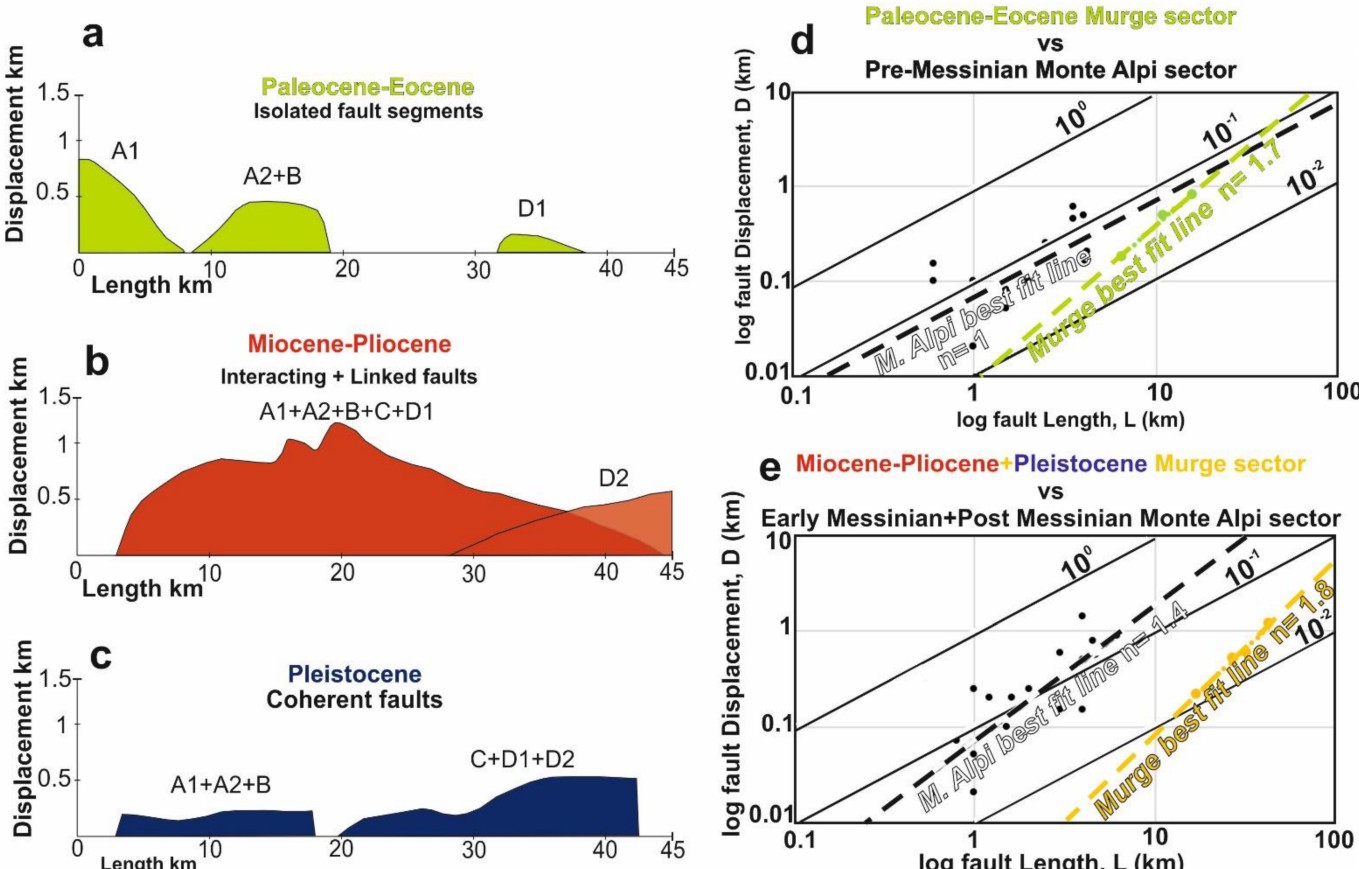

**Figure 11.** Fault displacement vs. fault length diagrams computed for (**a**) Paleocene–Eocene, (**b**) Miocene–Pliocene, and (**c**) Pleistocene times, the time intervals of fault growth investigated in this study. (**d**) Bi-logarithmic plot of displacement vs. length data computed for the Paleocene–Eocene faults of the study area (green points and dashed line) and for the Pre-Messinian faults of the Monte Alpi area (black points and dashed line, La Bruna et al. [36]. (**e**) Bi-logarithmic plot of displacement vs. length data computed for the Miocene–Pliocene and Pleistocene faults of the study area (yellow points and dashed line) and for the Early-Messinian and Post Messinian faults of the Monte Alpi area (black points and dashed line, La Bruna et al. [36].

We now consider the computed *n*-values to assess the occurrence of self-similar (*n* = 1) or scale-dependent (*n* ≠ 1) geometries for the study fault segments. The former geometries are indicative of a fault growth that occurred under constant driving stress conditions [80,94–96], whereas the latter ones could be due to the bias of data collection, and/or coexistence of brittle and plastic deformation mechanisms [29,81,90]. The results of our computations show that all *n*-values are above 1. In fact, the *n*-value obtained for the Paleocene–Eocene time interval is ca. 1.7 (Figure 11d), and those for both Miocene–Pliocene and Pleistocene time

intervals are ca. 1.8 (Figure 11e). The study fault zone was hence characterized by scale-dependent geometries throughout the whole Cenozoic era, although underestimation of fault length data due to poor resolutions at fault tips might have affected our interpretation of the seismic data [97,98]. Moreover, we highlight that the data scattering in the $D_{max}$-L diagrams was likely associated with the fault segment linkage processes discussed above.

The aforementioned *n*-values are quite similar to those computed for the post-Early Messinian fault network dissecting the Inner Apulian Platform exposed in the Monte Alpi area of southern Italy [36]. Accordingly, we interpret that the study fault zone crosscutting the Outer Apulian Platform grew, forming incipient scale-dependent geometries due to linkage processes among interacting isolated fault segments [26,78]. Focusing, therefore, on the time-dependent throw intervals computed for the study fault zone, our result differs from those documented by La Bruna et al. [36] for the Pre-Early Messinian high-angle faults of the Monte Alpi area and hence shows that the scale-dependent geometries characterized the study fault zone since the beginning of its activity. Moreover, following Walsh et al. [33], we suggest that the slightly steeper slopes of the Miocene–Pliocene and Pleistocene fault growth lines (Figure 11d,e) reflect the higher degree of maturity reached by the entire fault zone subsequent to Paleocene–Eocene deformation [28,86,98]. Finally, we invoke that the linkage processes took place in correspondence with the stepover/relay zones (Trudgill and Cartwright, 1994), as documented for the A2 + B fault segment during the Paleocene–Eocene time interval and for the A1 + A2 + B + C + D1 fault segment during the Miocene–Pliocene interval.

## 6. Conclusions

Aiming at assessing the Cenozoic fault growth evolution in the Outer Apulian Platform, we focused on a ca. 55 km-long extensional fault zone buried underneath the Plio-Quaternary foredeep deposits of the southern Apennines ftb, Italy. By analyzing public 2D seismic reflection profiles, well logs, and isochron maps data, we showed that, at present, the normal fault zone is made up of four individual fault segments. On the basis of the existing knowledge of both geometry and distribution of subsurface Eocene and Miocene sedimentary succession topping the Mesozoic Apulian carbonates, we computed both cumulative and incremental fault throw variations associated with the displacement of Top Cretaceous Top Eocene and Top Pliocene chrono-stratigraphic surfaces. The results of this computation were discussed in order to assess the possible similarities of the original fault segmentation with the present-day structural architecture of the study fault zone.

We documented that the original fault segments did not correspond to the four individual fault segments. Overall, the present-day structural configuration was interpreted as due to the growth, interaction, and linkage among interacting fault segments. Specifically, focusing on the computed values of incremental fault throw, the Paleocene–Eocene, Miocene–Pliocene, and Pleistocene time intervals were investigated. The Paleocene–Eocene faulting produced three distinct bell-shaped profiles, which were interpreted as due to the activity of isolated fault segments that dissected the Top Cretaceous chrono-stratigraphic surface. Differently, two bell-shaped throw profiles were computed for the Miocene–Pliocene time interval. The first one was interpreted as due to faulting of a ca. 40 km-long fault segment, the second one to a shorter, ca. 10 km-long fault segment. Considering the Pleistocene time interval, the computed throw profile showed the complete linkage among the individual fault segments and the that the whole fault zone behaved as a coherent fault system. We note that soft linkage processes took place during fault evolution at stepover/relay zones. According to the computed *n*-values for the single time intervals, we assessed that all computed *n*-values spanned between ca. 1.7 and 1.8 for all investigated time intervals, and these interpreted results were due to scale-dependent geometries associated with linkage processes among interacting fault segments, which determined a higher degree of maturity of the entire fault zone during evolution with time.

**Author Contributions:** Conceptualization: F.A., A.V.P. and V.L.B.; methodology: A.V.P., V.L.B. and G.P.; software: A.V.P. and G.P.; validation: A.V.P.; formal Analysis: F.A., A.V.P., V.L.B. and G.P.; investigation: A.V.P. and V.L.B.; resources: F.A. and G.P.; data Curation: A.V.P. writing—original draft preparation: F.A. and A.V.P.; writing—review and editing: F.A., A.V.P., V.L.B. and G.P.; visualization: A.V.P. and V.L.B.; supervision: F.A.; project administration: F.A.; funding acquisition: F.A. and G.P. All authors have read and agreed to the published version of the manuscript.

**Funding:** The research work was funded by the Reservoir Characterization Project (www.rechproject.com), an academic research project of the Basilicata and Camerino universities, Italy, funded by a consortium of energy companies.

**Acknowledgments:** The Petroleum Expert company is kindly acknowledged for the academic free license of the Move® software employed for this work. We thank the two anonymous reviewers for their thoughtful comments, which helped us to better elucidate the processes of data elaboration and interpretation. We also thank the editorial team, for the positive feedback and editorial work.

**Conflicts of Interest:** The authors declare no conflict of interest.

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
