# Peer review of "Cenozoic Fault Growth Mechanisms in the Outer Apulian Platform"

_geosciences, doi:10.3390/geosciences13040121_

Round 1

Reviewer 1 Report

Dear editor and authors,

The paper is clear and well written, relatively well respectful of the literature on the subject, and the observations and main interpretations appears to me correct and well linked. I therefore recommend publication after moderate revision considering two main points to consider in the discussion and some elements that could make figures clearer.

The first important point that needs to be discussed is that compared to the literature, you do not report cumulative displacement. The throws reported are during time intervals and the real fault length is unknown, i.e. it can be much larger than the length of the throw profiles observed during these intervals. I should therefore recommend clearly mentioning this as a main difference compared to the literature data in which displacement is the cumulative. I also recommend discussing the implication of this scale dependency identified in time intervals, for the scaling of cumulative displacement, at least for the sum of the three intervals inspected. Please also better consider the literature clearly reporting scale dependence and discussing its origin (see references below in the line comments).

The second point concerns negative throw on normal faults. First you must clarify what this is clearly (reverse component ?) and second find the possible explanations for this. I don’t really see how an unconformity can generate that and if so, the authors must clearly explain why. It would be easier to refer to normal fault inversion or thrusting prior to the observed normal faulting.

Elements concerning the figures as well as others very minor comments are listed below.

L51-64: It would be good to better clarify here the question that is addressed in this work with respect to the literature.

L71: This reference is good for scale independence for fracture and DB but not for faults.

I would suggest referencing Soliva and Schultz 2008 (Tectonics) instead, in which there is a review of the literature about scale invariance vs scale dependence for fault systems.

L83: There is no "c" in Figure 1.

Figure 1: I propose to change the color of the box (study area), in red for example, purple is yet used. This will allow to identify more quickly the study area.

L129: What is ftb?

L221: “n” has also been indirectly associated to mechanical properties. More precisely, contrasts of mechanical properties influencing the 3D shape of the faults.

Figure 4: Is it possible to get less shade on the surfaces to better see their 3D geometry?

Figure 5: Could you please give a map view as precise as possible of the fault scarp trace in order to see how much these segments are spatially distributed and linked?

Also, it would be better to put the white dashed line out of the fault scarp (a bit farther in the footwall). This will allow to better look at the scarp 3D morphology.

Figure 6: Please indicate fault letter shown in figure 5. I suggest adding letters for southwestern faults, also in figure 5.

L309: there is several bell shapes.

L338: Could you explain what is negative throw?

L316-319: Negative throw is apparent thrust? Then, don’t you think this is due to normal fault inversion or previous reverse faulting in a place where normal displacement is low. I am not sure that the unconformity alone can explain negative throw, and if so, please clarify how.

L397: Scale dependent is not only n>1 but instead n≠1. Most of the faults and fracture systems referred to be scale dependent are actually in the case of n<1.

L398: You must check but in my remind this theory applies for the growth of an isolated fault. This has to be recalled here if right. This is generally irrelevant since faults also grow by segment linkage.

L401: These papers does not relly focus on fault systems in which n differs from 1. I would suggest referencing: Cowie et al., 1994; Carbotte and Macdonald, 1994; Bohnenstiehl and Kleinrock, 2000; Schultz and Fossen, 2002; Soliva et al, 2005, 2006.

L411-415: Compared to the litterature, you do not report cummulative diplacement. The throws reported are during time intervals and the real fault length is unknown, i.e. it can be much larger than the length of the throw observed during these interval. I should therefore recommend clearly mentionning this as a differences compared to the litterature data. I also recommend discussing the implication of this scale dependency identified in time intervals, for the scaling of cummulative displacement, at least for the sum of the three intervals.

L461: Please, clarify here, as well as in the abstract, tha n is derived from throw calculated during time intervals. This is not cumulative displacement, which I believe is an important point to discuss.

Author Response

We thank the reviewer for the thoughtful and detailed comments. Our replies are reported in the annotated text uplodaed as attachment.

Reviewer 2 Report

Review for Geosciences

Title: Cenozoic Fault Growth Mechanisms in the Outer Apulian Platform

Authors: Fabrizio Agosta, Angela Vita Petrullo, Vincenzo La Bruna and Giacomo Prosser

General comments

The paper investigates the geometry and displacement of a 50 km-long segmented extensional fault zone in the fore-deep deposits of the southern Apennines, Italy, using public seismic data and well-logs. First, the authors provide a detailed geological setting, including the structural and stratigraphic settings. Next, the authors describe the geometry of the fault, followed by the thickness distribution of the Cenozoic deposits on the fault walls and associated displacement profiles. Finally, the authors discuss fault growth processes based on the observations.

I think that the topic of this manuscript is relevant to the journal. In particular, the paper provides insights into the deformation history and structural style of the studied area, which make the paper valuable for future regional studies. In addition, the paper describes an interesting example of a fault zone using appropriate methods and addresses the topical problem of fault development. However, since the results rely on data with relatively poor resolution, the scope of the paper in terms of a general understanding of fault growth remains relatively limited. Concerning the form, the paper is well-structured, and although I am not native English, I think it is well-written. Nevertheless, I have a few major and minor comments (see below), and I think the paper should be considered for publication after a major revision.

Main points:

Seismic and well-log data: As mentioned above, the paper deals with public seismic reflection data with relatively poor quality and well logs. While I think such data are relevant, the authors should discuss further, including quantitative insights, the uncertainty in their interpretations and results associated with such data. For example, the authors could discuss the minimum size of the structures they can observe based on their data, e.g. the minimum size of steps and the minimum size of faults, the uncertainty in the lateral correlation between segments or for calculating thickness distribution, as well as the uncertainty in fault length depending on the spacing of the profiles.

Seismic interpretation: The authors provide several interpreted profiles. However, sometimes their interpretation seems to cross-cut seismic reflectors. For example, such discrepancy can be seen on the footwall of the most Southern fault in Figure 3c and below 'Appulian platform' in Figure 6b. While this may have little impact on the final results, tidied-up interpretations should be more convincing for the readers.

Fault maps: While the paper provides data including raw profiles and horizons, I think that the paper will benefit from providing some fault maps. In particular, I found that the geometry of the fault traces and their segmentation is hard to see in Figures 5 and 7. From my point of view, simple contour maps showing the interpreted fault traces might be easier to read than 3D horizons with an oblique view. Otherwise, the authors could add the contours on the horizon, which is a straightforward task in Move.

Figures and references to Figures: I think the Figures could be cleaned up in terms of legends, orientations and scales. Most importantly, the authors often refer to the Figures in general and do not refer to the Figure subsets, like Figures 8b or 8c. Consequently, linking the text's description and the Figures is sometimes tricky. I will suggest the authors add some more explicit references to Figures and even annotate the figure to highlight the critical elements like the steps along the faults (L.246-255), the minor faults (L.259), or the depocenters (L.284-291).

Cumulative throw profiles: I found the use of the terms asymmetric, flat-shaped and bell-shaped profiles, which are specific terms used previously by other authors (Manighetti et al., 2001; 2005; Roche et al., 2012), for the throw profiles description is not clear. For example, the Cretaceous profile does not seem asymmetric but slightly decreasing southward, and the segments A and D of this profile are not clearly bell-shaped but relatively constant, with a 30% variation for A and a slight decrease southward for D. Overall, I will suggest the authors focus on the description of the data in detail and then try to fit a model. Otherwise, the term flat-shaped is used both in the abstract and discussion but not in the result section. Finally, what is causing the sudden changes in displacement at segment B? Is that an artefact? Is that a branch line?

Minor points:

L.49: Transtensional

L.76-79: I think that this should be deleted ?

Figure 1: There is no C in the figure and caption and no line for the cross-section (as indicated in the legend).

L.219: reference to Figure 10 before figures 4-9.

Figures 3a and 5: Maybe add (m) next to elevation.

L.284: This isn't very clear because Figure 8b shows the Miocene interval. And there is no map for the Pliocene interval.

L.284-291: The thickness values provided here do not fit with the scale bar in Figures 8b and c. Please clarify.

References

Manighetti, I., King, G. C. P., Gaudemer, Y., Scholz, C. H., & Doubre, C. (2001). Slip accumulation and lateral propagation of active normal faults in Afar. Journal of Geophysical Research: Solid Earth, 106(B7), 13667-13696.

Manighetti, I., Campillo, M., Sammis, C., Mai, P. M., & King, G. (2005). Evidence for selfsimilar, triangular slip distributions on earthquakes: Implications for earthquake and fault mechanics. Journal of Geophysical Research: Solid Earth, 110(B5).

Roche, V., Homberg, C., & Rocher, M. (2012). Fault displacement profiles in multilayer systems: from fault restriction to fault propagation. Terra Nova, 24(6), 499-504.

Author Response

(The authors gave the same response as above.)
